# Caffeine Protects Keratinocytes from *Trichophyton mentagrophytes* Infection and Behaves as an Antidermatophytic Agent

**DOI:** 10.3390/ijms25158303

**Published:** 2024-07-30

**Authors:** Diogo M. da Fonseca, Lisa Rodrigues, José Sousa-Baptista, Félix Marcos-Tejedor, Marta Mota, Rodrigo A. Cunha, Chantal Fernandes, Teresa Gonçalves

**Affiliations:** 1FMUC—Faculty of Medicine, University of Coimbra, Rua Larga, 3004-504 Coimbra, Portugal; dmfonseca@igc.gulbenkian.pt (D.M.d.F.); josepedrobaptista1237@gmail.com (J.S.-B.); marta.mota83@gmail.com (M.M.); cunharod@gmail.com (R.A.C.); 2CNC-UC—Center for Neuroscience and Cell Biology of Coimbra, University of Coimbra, 3004-504 Coimbra, Portugal; lisa1cor@gmail.com (L.R.); xantal@gmail.com (C.F.); 3CIBB—Centre for Innovative Biomedicine and Biotechnology, University of Coimbra, 3004-504 Coimbra, Portugal; 4Department of Medical Sciences, Faculty of Health Sciences, University of Castilla-La Mancha, 45600 Talavera de la Reina, Toledo, Spain; felix.marcostejedor@uclm.es

**Keywords:** caffeine, antidermatophytic, dermatophytes, *Trichophyton mentagrophytes*

## Abstract

Caffeine affords several beneficial effects on human health, acting as an antioxidant, anti-inflammatory agent, and analgesic. Caffeine is widely used in cosmetics, but its antimicrobial activity has been scarcely explored, namely against skin infection agents. Dermatophytes are the most common fungal agents of human infection, mainly of skin infections. This work describes the in vitro effect of caffeine during keratinocyte infection by *Trichophyton mentagrophytes*, one of the most common dermatophytes. The results show that caffeine was endowed with antidermatophytic activity with a MIC, determined following the EUCAST standards, of 8 mM. Caffeine triggered a modification of the levels of two major components of the fungal cell wall, β-(1,3)-glucan and chitin. Caffeine also disturbed the ultrastructure of the fungal cells, particularly the cell wall surface and mitochondria, and autophagic-like structures were observed. During dermatophyte–human keratinocyte interactions, caffeine prevented the loss of viability of keratinocytes and delayed spore germination. Overall, this indicates that caffeine can act as a therapeutic and prophylactic agent for dermatophytosis.

## 1. Introduction

Caffeine (1,3,7-trimethylxanthine), a purine alkaloid, is the most widely consumed psychostimulant, found in several beverages such as coffee, tea, cocoa, yerba mate, guarana, or cola nuts. Several beneficial effects for human health have been associated with caffeine intake such as increasing cognitive and memory performance and increasing healthspan upon aging. Its recognized bioactive properties include being anti-inflammatory, analgesic, and anti-tumoral [1,2,3]. Several studies show that the consumption of caffeine, thanks to its antioxidative properties, prevents the occurrence or progression of chronic diseases, where reactive oxygen or nitrogen species are involved [3,4]. Recently, a comprehensive review addressed the effects of caffeine on human health, as well as delivery formulations to enhance the bioavailability of caffeine [3]. Caffeine and its derivatives are also widely used in cosmetics. The cosmetic antioxidant properties of caffeine make it an important protective and preventative agent against UV damage radiation and photoaging of the skin [5,6]. This led us to hypothesize that caffeine might also be beneficial during skin infection since the antibacterial and antifungal activity of caffeine has been previously reported [7,8,9,10,11]. Caffeine has also been described as interfering in fungal cell growth and is widely used in in vitro assays aiming to study the mechanisms regulating cell wall synthesis [12,13,14,15] and, more recently, the mechanisms of antifungal resistance [16]. Previously, we have described the antifungal effect of spent coffee grounds against dermatophytes, a group of fungi responsible for skin infections [17]. However, the antifungal properties of caffeine on the dermatophytes involved in human skin infection have not been unraveled. Dermatophytes are filamentous fungi with a tropism for keratinized structures and are an important cause of skin, nail, and hair infections, generically designated as dermatophytosis. Although these infections are not life-threatening, they can significantly affect the patient’s quality of life. The management of these infections has become an important public health issue, due to the incidence of recurrent, recalcitrant, or extensive infections [18]. The current treatment for dermatophytosis comprises topical application of antifungal agents, while administration of oral antifungals is indicated for more extensive infections, although they have limited effectiveness and toxicity [18,19].

This work reports the susceptibility of *T. mentagrophytes* to caffeine and the impact of caffeine in the progression of the in vitro infection of HaCaT human keratinocytes with this dermatophyte, together with the evaluation of its cytotoxicity, fungal spores’ germination, and cell viability upon infection. The antifungal mechanism of action was approached by quantification of the main fungal cell wall components, chitin and β-1,3-glucan, and how caffeine affects the germination of fungal spores during in vitro infection. 

## 2. Results

### 2.1. Antidermatophytic Activity of Caffeine

Antifungal susceptibility testing was performed to determine the minimal inhibitory concentration (MIC) of caffeine, following the EUCAST E.DEF 9.3.1 standards for filamentous fungi. The antifungal susceptibility assays showed that caffeine is an antidermatophytic with a MIC value of 8 mM for *T. mentagrophytes*. 

### 2.2. Modulation of Fungal Cell Wall β-1,3-Glucan and Chitin in Response to Caffeine

To approach the mechanism of action of caffeine, we determined the modulation of two cell wall components of *T. mentagrophytes* as a model organism. The results showed that all the tested concentrations of caffeine caused a significant decrease in the levels of the cell wall β-(1,3)-glucan in *T. mentagrophytes* (Figure 1A).

The quantification of chitin in *T. mentagrophytes* grown in media supplemented with several concentrations of caffeine, showed that different concentrations of caffeine affected differently the chitin cell wall content. While 1 mM of caffeine did not change the chitin cell wall levels, 5 mM and 10 mM concentrations of caffeine significantly increased the levels of this cell wall component of *T. mentagrophytes* (Figure 1B). At the 10 mM caffeine level, the highest concentration tested, a roughly threefold increased production of chitin occurred when compared with control conditions; this indicates that high concentrations of caffeine lead to an upregulation of chitin synthesis by this dermatophyte. 

### 2.3. Characterization of Ultrastructural Changes

Transmission electron microscopy was used to appraise the ultrastructural changes in the hyphae of the fungi grown in caffeine-containing media. The morphology of *T. mentagrophytes* grown in the absence of caffeine (Figure 2A–D) showed fungal hyphae with a regular clear cytoplasm and a homogeneous cell wall with a smooth surface. One of the most marked features of the ultrastructure of fungal cells is a high number of mitochondria, with a normal morphology characterized by well-defined cristae and regular nuclei. Figure 2A,C,D shows the presence of large vacuoles with denser electronegative compounds inside the control fungal cells, probably representing reserve materials required for hyphal growth. 

In the presence of 10 mM caffeine (Figure 2E–L), abnormal cellular morphologies and ultrastructure modifications were observed when compared with the control (Figure 2A–D). The clearest modification occurs in the cell wall surface, with a rougher surface and less homogeneous structure in fungi grown in 10 mM caffeine compared with control cells. The stability of the cell wall structure seems compromised since there were materials detaching from it (as seen in the central cell in Figure 2L). Interestingly, in Figure 2H, the chromatin seems to be migrating to opposite poles of the nucleus, which suggests that the fungal cell is undergoing mitosis and consequently dividing. Another cellular structure that exhibits modifications in the presence of caffeine is the mitochondria. Strikingly, the ultrastructure of these organelles showed an aberrant morphology with abnormal cristae (see zoomed crops in Figure 2G–J). Signs of autophagy can be observed in Figure 2F,K with the characteristic membrane whorls. Another aspect worth noting is that the extracellular medium of fungi incubated with caffeine is full of debris, not observed in the extracellular milieu of fungi grown in control conditions. Measurement of the thickness of *T. mentagrophytes* cell walls did not reveal significant alterations when grown in the presence and absence of caffeine (results not shown). 

### 2.4. Antifungal Properties of Caffeine during Keratinocyte Infections by T. mentagrophytes Spores

#### 2.4.1. Viability of Keratinocytes upon Infection with *T. mentagrophytes* Spores

To understand how caffeine affects the viability of HaCaT cells, control assays were performed by incubating keratinocytes with the selected caffeine concentrations, without the addition of fungal microconidia (Figure 3A).

During the interaction assay period (12 h), there were no differences in the viability of keratinocytes when they were incubated in the presence of 50 μM caffeine; however, viability decayed in the presence of 1 mM, 5 mM, and 10 mM caffeine (Figure 3A). Although these results show that increasing concentrations of caffeine have a negative effect on the viability of HaCaT cells, the viability of the cells under all the conditions was above 70% (Figure 3A). A completely different scenario was observed when HaCat cells were infected with *T. mentagrophytes* (Figure 3B). In these conditions, both without or in the presence of 50 μM and 1 mM of caffeine, the viability of HaCaT cells dramatically decreased to 11% during the 12 h infection period. However, this decline in cell viability was attenuated by the presence of 5 mM and 10 mM caffeine concentrations to values of 27% and 56%, respectively (Figure 3B). This indicates that caffeine prevents the loss of viability of keratinocytes during the course of *T. mentagrophytes* infection. 

#### 2.4.2. Ungermination of Microconidia during HaCaT Infection

These assays were performed to understand how caffeine impacts dermatophyte microconidia germination during the course of an in vitro keratinocyte infection. The results obtained show that 10 mM caffeine reduced *T. mentagrophytes* spore germination during infection of human keratinocytes (Figure 4).

The presence of caffeine led to a higher number of ungerminated conidia. This effect was more evident at 9 h of infection, with a 460% increase in the number of ungerminated microconidia (Figure 4). 

## 3. Discussion

Caffeine is a secondary metabolite present in over 100 plant species [8]. Although only a few papers report the antifungal properties of caffeine, its antidermatophytic activity was not known. We previously described that spent coffee ground extracts were endowed with antifungal properties, including antidermatophytic activity [17]. This was observed for both caffeinated and decaffeinated spent ground coffee extracts: the differences between the two extracts were minimal, but both had antidermatophytic effect. However, in that previous study, the components of the extracts were not isolated, so the effect was ascribed to a complex mixture of phytochemicals [17]. The main goal of the present work was to unveil the particular antidermatophytosis properties of caffeine.

The in vitro antidermatophytic effect of caffeine was determined using a microdilution broth, following EUCAST standards, on *T. mentagrophytes* with a MIC of 8 mM. These results are in good agreement with previous studies on the yeast *Candida albicans* that reported a MIC of caffeine of 12.5 mM [20]. Other studies also in *C. albicans* showed a different MIC of caffeine of 0.1 mM (25 mg/L) [21] using caffeine extracted from tea and 150 mM (30 mg/mL) [22] using pure caffeine. Comparison with filamentous fungi is limited because data are scarce and the few available data were obtained with different methodologies [23].

As an attempt to unravel the therapeutic target of caffeine, we quantified the modulation of β-1,3-glucan and chitin contents of *T. mentagrophytes* grown in the absence and presence of caffeine. The β-(1,3)-glucan and chitin were selected because, apart from being two structurally important polysaccharides present in this cellular structure and strongly accounting for its integrity, they are also important pathogen-associated molecular patterns (PAMPs) to consider when studying infection and inflammation phenomena triggered by these fungi [24,25,26]. It was observed that there was a decrease in β-(1,3)-glucan levels at all tested concentrations of caffeine together with an increase in chitin cell wall contents upon exposure to 5- and 10-mM caffeine in relation to the control. Some fungi also increase chitin levels in response to echinocandins as a salvage mechanism to compensate for the decreased contents of β-glucan [27,28,29,30]. Previously, we also reported some fluctuations of the chitin and/or β-glucan cell wall content in dermatophytes by the action of natural extracts, but we never observed such a clear paradoxal effect associated with the decreased β-glucan levels [31,32]. This may indicate that the decreased levels of β-glucan lead to a marked cell wall destabilization or that caffeine activates the chitin cell wall synthesis. Although few studies report the antifungal properties of caffeine, caffeine has been described as interfering with the growth of fungal cells and it is widely used in in vitro assays studying the mechanisms of the regulation of cell wall synthesis [13,14]. Recently, it was proposed that caffeine might be an “epigenetic drug” reducing antifungal resistance [16]. Caffeine is an inhibitor of cAMP phosphodiesterase and stimulates a dual phosphorylation of ScSlt2, the MAP kinase component of the PKC cell wall integrity signal transduction pathway [33]. For example, the growth of *C. albicans* in a medium containing 12 mM of caffeine resulted in a significantly elevated expression from all CHS promoters [14]. Therefore, we can speculate that the two-to-three-fold increase in chitin cell wall levels now observed upon exposure to 5 mM and 10 mM of caffeine might also be due to the activation of chitin synthesis.

The impact of caffeine on the ultrastructure of *T. mentagrophytes* was evaluated by TEM and showed profound changes in response to caffeine. The surface of the cell wall was modified in the presence of caffeine and presented a rougher aspect with loosening materials to the extracellular milieu. This ultrastructural modification might be due to changes in the cell wall components as observed with other fungi treated with inhibitors of glucan synthesis, of chitin synthase, or of melanin synthesis [34]. It was found that mitochondria were scarce in caffeine-grown fungi when compared with the control and displayed an irregular morphology and abnormal cristae. In contrast, in humans, it is described that caffeine promotes mitochondrial biogenesis and improves mitochondrial functionality, playing an important protective role against oxidative stress [35,36,37]. The TEM study also showed that *T. mentagrophytes* grown in the presence of 10 mM caffeine had irregular structures, which appeared rolled in whorls. These abnormal structures were reported before as signs of autophagy [38]. In fact, we previously observed similar modifications with extracts of spent ground coffee and with other natural extracts [17,31,32]. Vacuoles play an important role in homeostasis and in the maintenance of a balanced chemical composition of the cytoplasm in the face of fluctuating external conditions; one of their mechanisms is the excreting of water into the cell to compensate for an unbalanced turgor pressure [39]. Since these structures appeared in higher numbers and some of them were abnormally large, it might be a mechanism of *T. mentagrophytes* to compensate for changes in the osmotic equilibrium due to a more fragile cell wall. 

One of the most important objectives of this work was to study the impact of caffeine, a drug used in cosmetics, during the course of a (skin) fungal infection. For this, we studied the interaction of *T. mentagrophytes* spores with human keratinocytes (HaCaT cell line). Keratinocytes constitute most of the epithelial cells in human skin [40]. These cells are responsible for the formation of a natural barrier against physical, chemical, and microbial aggressions. Keratinocytes detect and respond to stimuli producing immune-inflammatory mediators that in turn trigger a more specific response by cells of the immune system, some of which are recruited to the site where the aggression was detected [41]. Keratinocytes also play an important role in wound healing [42]. Here, it was explored whether caffeine could interfere with the damage of human keratinocytes during the infection of *T. mentagrophytes*. It was observed that the infection of a HaCaT cell culture with spores of *T. mentagrophytes* during 12 h led to a decrease in cell viability to 11% of the control non-infected cells. The exposure to 10 mM caffeine during this 12 h interaction period prevented the loss of viability of keratinocytes. Surprisingly, caffeine alone led to a decrease in keratinocyte viability, although remaining above 70% with all the tested concentrations of caffeine. In other stressful conditions, it was described that 5 mM caffeine reduces HaCaT cell viability and promotes apoptosis upon exposure to ultraviolet B radiation [2]. It was also reported that this concentration of caffeine prevents human epidermal keratinocyte (HEK cell line) proliferation and migration, therefore suggesting that it may have an inhibitory effect in wound closure and epithelization during in vitro wound assays. The authors hypothesized that this delay, caused by caffeine in cell migration and epithelization, might likely be due to alterations of the cytoskeleton caused by caffeine, but the underlying mechanism remains to be elucidated [43]. We now describe for the first time that, during the course of an infection (in this case by *T. mentagrophytes*)*,* the presence of caffeine-protected keratinocytes, preserving their viability. This might be explained by the antioxidant effect of caffeine since under conditions of infection, oxidative stress increases dramatically and this is usually also deleterious for the host cell, although contributing to killing the pathogen. Moreover, upon the infection of keratinocytes with a dermatophyte in the presence of caffeine, there was a decay in the germination of fungal spores. Undoubtfully, these two aspects together indicate an alleviation of the infection process since the ability of a fungal infection such as tinea to proceed and succeed depends on the germination of spores. 

The most important conclusions that were revealed by the present study are that caffeine, especially at a concentration of 10 mM, a fifteen times lower concentration than the one found in most commercially available topical solutions (3%; [6]), inhibited *T. mentagrophytes* growth. Caffeine also altered both β-(1,3)-glucan and chitin, two important structural components of the fungal cell wall required for the robustness of the fungus, for an efficient recognition of this pathogenic microorganism by target host cells and for triggering an efficient immune response to promote its clearance during an infection. It was also shown that caffeine preserved the viability of keratinocytes during an in vitro infection insult and delayed microconidia germination. Considering the unmet medical need for more effective antifungal chemotherapeutic agents or approaches for treating dermatophytosis, natural products have been a successful source for the discovery of new drugs [44,45,46,47]. Therefore, the present work is an opportunity for the use of topical formulations containing caffeine, opening novel perspectives for the implementation of a clinical trial to test the topical use of caffeine as a novel therapeutic approach to dermatophytosis.

## 4. Materials and Methods

### 4.1. Fungal Strains and Culture Conditions

*T. mentagrophytes* was the model dermatophyte selected. This fungus was kindly provided by Professor Carmen Lisboa of the Laboratory of Microbiology, Faculty of Medicine, University of Porto. Fungi were cultured at 30 °C in Potato Dextrose Agar medium (PDA, BD Biosciences^®^, San Jose, CA, USA). 

Fungal spores were harvested from solid cultures, submerging the mycelial mat with 0.1% Tween 80 solution (*v*/*v*, Sigma-Aldrich^®^, St. Louis, MO, USA). Then, the suspensions of spores were filtered through sterile handmade sacred linen filtration systems to remove hyphae fragments. The filtered suspension was washed twice with Phosphate Buffered Saline (PBS) (10 mM Na_2_PO_4_, 1.8 mM KH_2_PO_4_, 137 mM NaCl, 2.7 mM KCl (*w*/*v*), pH 7.3, by centrifugation at 16,060× *g* for 10 min, at 4 °C, and the pellets were resuspended in PBS. After another washed step under the same conditions, the pellet was resuspended either in PBS and diluted in RPMI R1383 (Sigma-Aldrich^®^, St. Louis, MO, USA) for posterior use for susceptibility assays or in YME (0.4% yeast extract (*w*/*v*), 1% glucose (*w*/*v*), 1% malt extract (*w*/*v*)) for liquid cultures or in DMEM D5648 (Sigma-Aldrich^®^, St. Louis, MO, USA) supplemented with 10% non-inactivated Fetal Bovine Serum, 10 mM HEPES, 12 mM NaHCO_3_, and 2 mM L-glutamine for later use in infection assays. Concentration of spores in the purified suspensions was estimated in microconidia/mL using a hemocytometer. 

For the quantification assays of fungal cell wall components and for transmission electron microscopy (TEM) analysis, 2 × 10^5^ fungal microconidia were inoculated in 100 mL of liquid YME supplemented with caffeine (Sigma-Aldrich^®^, St. Louis, MO, USA) at concentrations of 1, 5, or 10 mM. Caffeine was added to the sterile media by aseptically adding 10 mL of each caffeine sterile stock solution (*w*/*v*). Cultures were incubated at 30 °C with constant orbital shaking at 120 rpm from 3 up to 25 days upon inoculation (control cultures without caffeine and cultures supplemented with 1 mM caffeine were grown for 3 days, 5 mM caffeine cultures for 9 days, and 10 mM cultures for 25 days to enable fungal growth).

### 4.2. Antidermatophytic Activity of Caffeine

The minimum inhibitory concentration (MIC) causing the inhibition of the growth of the selected dermatophytes was determined by microdilution, as previously described [17,32], following the E.DEF 9.3.1 EUCAST standards. The MIC was defined as the lowest concentration for which caffeine inhibited fungal growth.

### 4.3. Fungal Cell Wall β-1,3-Glucan and Chitin Quantifications

The quantification of β-1,3-glucan levels in fungal cell walls was performed using the aniline blue assay, as described before [47]. Mycelia were sonicated in 1 M NaOH, and the β-1,3-glucan concentration was determined by aniline blue fluorescence at 405 nm excitation and 460 nm emission in a fluorimeter (Spectra Max^®^ ID3, Molecular Devices, San José, CA, USA) [47]. 

The quantification of chitin in the cell wall was performed by measuring the glucosamine released by acid hydrolysis of purified cell walls, as described before [31]. The absorbance was read at 520 nm on a plate reader using a SpectraMax^®^ Plus 384 spectrophotometer (Molecular Devices, San Jose, CA, USA).

### 4.4. Characterization of Morphological and Ultrastructural Changes

The ultrastructural changes induced by caffeine in *T. mentagrophytes* were analyzed by transmission electron microscopy (TEM). To perform TEM analysis, the inoculum was cultured in liquid medium with 10 mM caffeine or with no caffeine (control cultures). Samples were washed and then fixed with 2.5% glutaraldehyde in 0.1 M sodium cacodylate buffer (pH 7.2) for 2 h. Fixation was performed as described previously [31]. Observations were carried out on a FEI-Tecnai^®^ G2 Spirit Bio TwinTM transmission electron microscope at 100 kV. 

### 4.5. Infection Assays

The HaCaT cell line of immortalized human keratinocytes was obtained from the German Cancer Research Centre (Heidelberg, Germany). Cells were maintained in Dulbecco’s Modified Eagle Medium D5645 (Sigma-Aldrich^®^, St. Louis, MO, USA) supplemented with 10% non-inactivated Fetal Bovine Serum (FBS), 10 mM HEPES, 12 mM sodium bicarbonate, and 2 mM L-glutamine (DMEM), at 37 °C in 5% CO_2_ atmosphere_,_ until reaching 70% confluency. All infection assays were performed using HaCaT cells from passages #35 to #55. For the in vitro infection assays, HaCaT cells were trypsinized (trypsin-EDTA solution; Sigma-Aldrich^®^, St. Louis, MO, USA) and, depending on the assay type, 1 to 2 × 10^5^ keratinocytes/well were seeded in 12-multiwell plates. HaCaT cells were allowed to sediment and adhere overnight at 37 °C in a 5% CO_2_ atmosphere and then incubated with *T. mentagrophytes* microconidia at a multiplicity of infection (MOI) of 1:1. To assess the impact of caffeine in keratinocyte–microconidia interaction, caffeine solutions were also added at different concentrations (50 μM, 1 mM, 5 mM, or 10 mM). 

### 4.6. Keratinocyte Viability Assay

HaCaT cell viability was assessed using an MTT assay, according to manufacturer’s instructions. Briefly, 12-well keratinocyte plates were incubated with *T. mentagrophytes* microconidia and different caffeine concentrations for 6, 9, and 12 h. After each incubation period, media was discarded and replaced by DMEM with MTT solution (Sigma-Aldrich^®^, St. Louis, MO, USA), at a final concentration of 500 μg/mL per well. Plates were incubated again for 2 h to allow keratinocytes to metabolize the compound, and dimethyl sulfoxide (DMSO; Sigma-Aldrich^®^, St. Louis, MO, USA) was then added to each well. The contents of each well were transferred to 96-well microtiter plates in 100 μL triplicates and absorbance at 570 nm was measured in a SpectraMax^®^ PLUS 384 spectrophotometer. 

### 4.7. Ungerminated Microconidia Assays

HaCaT cells and *T. mentagrophytes* microconidia were co-incubated, as described above, and, after each incubation period, the plates were transferred to ice. The wells were immediately scraped and centrifuged at 16,060× *g* for 10 min at 4 °C. After centrifugation, supernatants were rejected, and the pellets were resuspended and homogenized in ice-cold PBS. The ungerminated microconidia were counted using a hemocytometer. 

### 4.8. Statistical Analysis

Statistical analysis was performed using GraphPad^®^ Prism 5 software (GraphPad Software, Inc., La Jolla, CA, USA). Data are presented as mean ± standard error of the mean (SEM), and significance values are presented as *p* < 0.05, *p* < 0.01, or *p* < 0.001 (*, **, and ***, respectively). 

## 5. Conclusions

The main conclusions are the following: (1) caffeine, especially at 10 mM, a concentration fifteen times lower than that found in commercially available topical solutions, inhibits the growth of the dermatophyte *T. mentagrophytes*; (2) caffeine modifies both β-(1,3)-glucan and chitin, reducing the robustness of the cell wall of the dermatophyte; and (3) caffeine prevents keratinocyte cell death during an in vitro infection by *T. mentagrophytes* and delays microconidia germination. 

The present work opens novel perspectives for the implementation of a clinical trial to test the topical use of caffeine as a novel therapeutic approach to dermatophytosis.

## Figures and Tables

**Figure 1 ijms-25-08303-f001:**
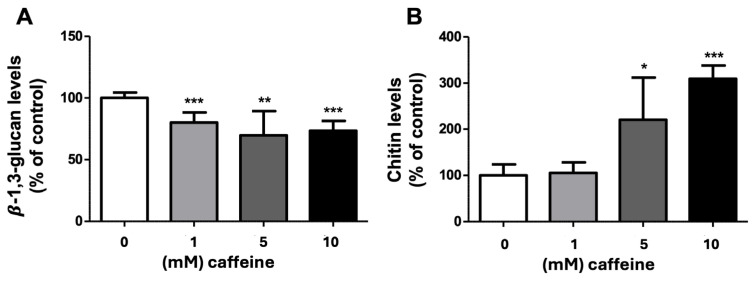
Effect of caffeine on β-(1,3)-glucan and chitin levels of *T. mentagrophytes* cell wall. *T. mentagrophytes* microconidia were inoculated in YME supplemented with 1, 5, or 10 mM of caffeine prior to the quantification of (**A**) β-(1,3)-glucan or (**B**) chitin levels in the fungal cell wall. Under the control condition, the fungus was grown in YME containing no caffeine. Results are presented as mean ± SEM (*n* = 3); * *p* < 0.05, ** *p* < 0.01, *** *p* < 0.001 with an unpaired *t*-test.

**Figure 2 ijms-25-08303-f002:**
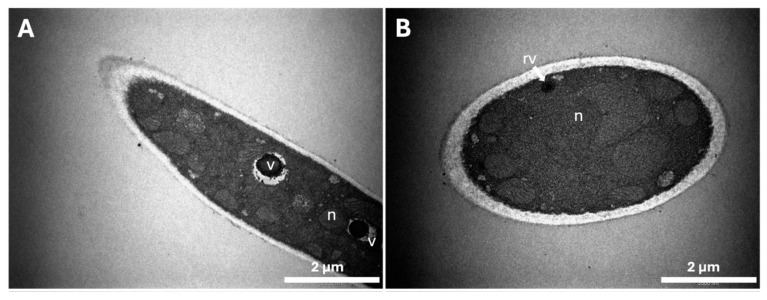
Impact of caffeine on *T. mentagrophytes* ultrastructure. Fungal microconidia were inoculated in YME in the absence or presence of 10 mM caffeine. Acquired images from (**A**–**D**) represent fungi grown in the absence of caffeine, and images from (**E**–**L**) represent fungi grown in media supplemented with 10 mM caffeine. “v” identifies vacuoles, “n” nuclei.

**Figure 3 ijms-25-08303-f003:**
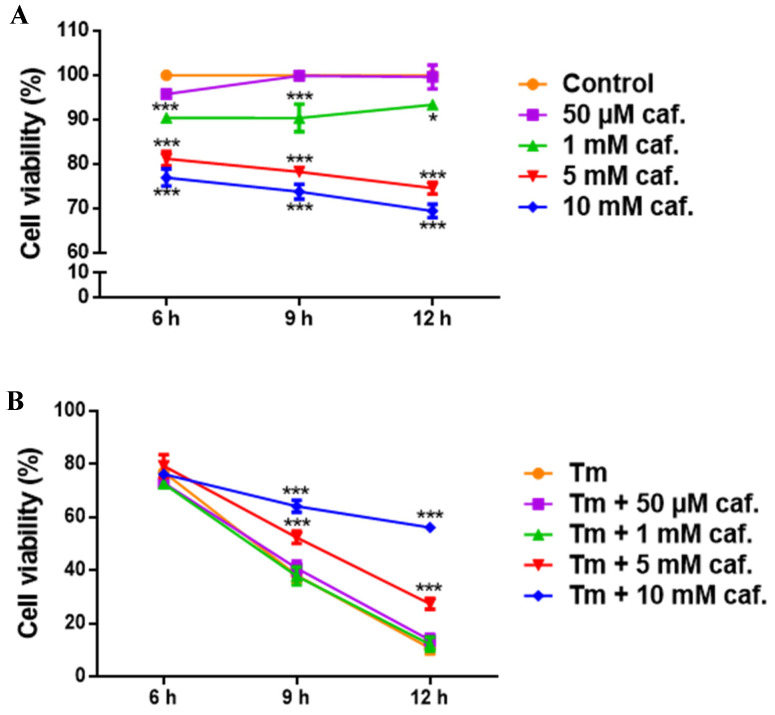
Effect of caffeine on HaCaT cell viability and impact during an in vitro infection with *T. mentagrophytes* (Tm) microconidia. Keratinocytes were exposed to caffeine (caf.) at different concentrations for 6, 9, and 12 h. Cell viability was quantified using the MTT assay. Under control condition, no caffeine was added to cells. (**A**) The viability of HaCaT cell was determined in the absence of fungal microconidia and (**B**) in the presence of fungal *T. mentagrophytes* spores. Results are presented as mean ± SEM (*n* = 3); * *p* < 0.05, *** *p* < 0.001 (vs. control in (**A**) and vs. Tm in (**B**)) using two-way ANOVA.

**Figure 4 ijms-25-08303-f004:**
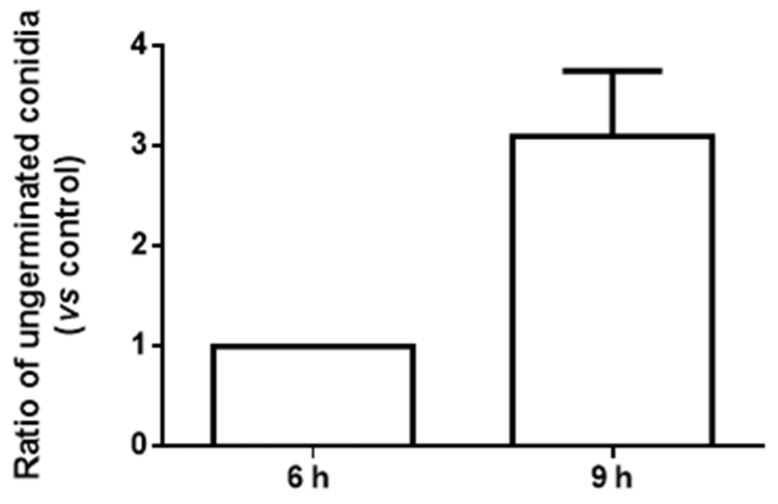
Effect of caffeine on *T. mentagrophytes* microconidia germination during the course of an in vitro infection. HaCaT cells were infected with *T. mentagrophytes* microconidia in the absence (control condition) or in the presence of 10 mM caffeine. After each incubation period, the number of ungerminated microconidia was counted. The number of ungerminated microconidia in control conditions (no caffeine) was normalized to 1. Results are presented relative to control as mean ± SEM (*n* = 3).

## Data Availability

Data will be available upon request.

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
