# Peer review of "Caffeine Protects Keratinocytes from Trichophyton mentagrophytes Infection and Behaves as an Antidermatophytic Agent"

_ijms, 2024, doi:10.3390/ijms25158303_

Round 1
Reviewer 1 Report
Comments and Suggestions for Authors
Authors have presented the protective effect of Caffeine on keratinocytes from Trichophyton mentagrophytes infection. The concept of MS is good. However, the manuscript needs extensive revision and rewriting of sentences for better understanding.
The specific comments, which could help to improve the manuscript, are:
1. The manuscript should be revised for grammatical & punctuation errors.
2. Line 29: which current treatment can be associated with adverse effects and toxicity.
3. Line 34: psychoactive stimulant?
4. Line 52-67: rewrite to make it meaningful.
5. Line 314: as described before [31] [31Fer- 314 nandes et al., 2023].
6. Figure 3A & B: Better to present each test group with different color lines for better interpretation.
7. Provide the figures as supplementary material for Ungermination of microconidia during HaCaT infection (2.4.2.). Specially for ungerminated microconidia counting using a haemocytometer.
8. It is better to add conclusion section showing highlights of the research.
Comments on the Quality of English LanguageExtensive editing of English language is required
Author Response
Authors have presented the protective effect of Caffeine on keratinocytes from Trichophyton mentagrophytes infection. The concept of MS is good. However, the manuscript needs extensive revision and rewriting of sentences for better understanding.
The specific comments, which could help to improve the manuscript, are:
- The manuscript should be revised for grammatical & punctuation errors.
One of the authors is proficient in English with over 300 papers published, ensuring that the quality of the English should be appropriated to vehiculate the message. The manuscript was carefully revised by this author.
- Line 29: which current treatment can be associated with adverse effects and toxicity.
We thank the reviewer for this comment. We agree that this sentence makes no sense at the end of the abstract so we erased it.
- Line 34: psychoactive stimulant?
The terminology was correct, replacing ‘psychoactive stimulant’ by ‘psychostimulant’.
- Line 52-67: rewrite to make it meaningful.
Instead of:
Before we described the antifungal effect of spent coffee ground against dermatophytes, a group of fungi responsible for skin infections [17] but the antifungal properties of caffeine on the dermatophytes human skin infection have yet to be unraveled. These filamentous fungi infect keratinized structures, inducing superficial infections of the human skin. Although these infections are not life-threatening, they may significantly affect the patient quality of life. Their management have become an important public health issue, due to incidence of recurrent, recalcitrant or extensive infections [18]. Current treatments for localized infections caused by these fungi comprise topical application of antifungal agents, while administration of oral antifungals with limited effectiveness and toxicity is indicated for more extensive infections, [18,19].
the sentence now reads:
Previously, we have described the antifungal effect of spent coffee ground against dermatophytes, a group of fungi responsible for skin infections [17]. However, the antifungal properties of caffeine on the dermatophytes involved in human skin infection have not been unraveled. Dermatophytes are filamentous fungi with a tropism for keratinized structures and are an important cause of skin, nails and hair infections, generically designated as dermatophytosis. Although these infections are not life-threatening, they can significantly affect the patient’s quality of life. The management of these infections has become an important public health issue, due to the incidence of recurrent, recalcitrant or extensive infections [18]. The current treatment for dermatophytosis comprises topical application of antifungal agents, while administration of oral antifungals is indicated for more extensive infections, although they have limited effectiveness and toxicity [18,19].
- Line 314: as described before [31] [31Fer- 314 nandes et al., 2023].
Changed accordingly.
- Figure 3A & B: Better to present each test group with different color lines for better interpretation.
The figure was modified according to the reviewer’s suggestion.
- Provide the figures as supplementary material for Ungermination of microconidia during HaCaT infection (2.4.2.). Specially for ungerminated microconidia counting using a haemocytometer.
As explained in the Materials and Methods’ section, fresh preparations of the co-cultures were observed and counted immediately in the haemocytometer. Therefore, we do not have any representative microscopic images of these preparations.
It is better to add conclusion section showing highlights of the research.
Our understanding is that the Conclusion section is optional when the Discussion is unusually long or complex, as indicated in the ‘Instructions to Authors’, whereas there are no limitations related to the Discussion section. Therefore, the last paragraph of the Discussion Section summarized the main results and conclusions of the work in the previous version of the manuscript, which would make redundant the inclusion of a Conclusions section. However, since the reviewer specifically asked for this additional section, we are glad to include such section, as follows (Line 543):
- Conclusions
The main conclusions are that: 1) caffeine, especially at 10 mM, a concentration fifteen times lower than that found in commercially available topical solutions, inhibits the growth of the dermatophyte T. mentagrophytes; 2) caffeine modifies both β-(1,3)-glucan and chitin, reducing the robustness of the cell wall of the dermatophyte; 3) caffeine prevents keratinocytes cell death during an in vitro infection by T. mentagrophytes and delays microconidia germination.
The present work opens novel perspectives for the implementation of a clinical trial to test the topical use of caffeine as a novel therapeutic approach to dermatophytosis.
Comments on the Quality of English Language
Extensive editing of English language is required
As explained before, one of the authors is proficient in English with over 300 papers published, ensuring that the quality of the English should be appropriated to vehiculate the message. The manuscript was carefully revised by this author.
Reviewer 2 Report
Comments and Suggestions for Authors
The manuscript presents studies on the effect of caffeine as an antidermatophyte agent. It was shown that caffeine can be used in the treatment of dermatophytosis, especially caused by Trichophyton mentagrophytes.
The manuscript is interesting and well-written, but needs some minor revisions.
1. Figure 3 - I think that instead of lines labeled with different symbols it would be better to label them with different colors. This would definitely increase the readability of Part B in particular.
2. Figure 4 and the commentary on it - the statement that there was a 460% increase does not sound very good. It would be better to normalize the number of ungerminated microconidia under control conditions (without caffeine) to, for example, a value of 1.
3. Lines 135, 143 - instead of HaCat it should be HaCaT
4. Lines 285-288 - This passage needs to be corrected. First, there is no information on what the suspension was washed with, while the information in parentheses refers to centrifugation conditions. Second, the information about washing and suspending the precipitate in PBS solution is repeated twice.
5. Lines 314-315 - "[31Fernandes et al., 2023]" should be removed.
6. Lines 318, 336, 352 - T. mentagrophytes should be written in italics.
Author Response
The manuscript presents studies on the effect of caffeine as an antidermatophyte agent. It was shown that caffeine can be used in the treatment of dermatophytosis, especially caused by Trichophyton mentagrophytes.
The manuscript is interesting and well-written, but needs some minor revisions.
- Figure 3 – I think that instead of lines labeled with different symbols it would be better to label them with different colors. This would definitely increase the readability of Part B in particular.
Figure 3 was changed accordingly.
- Figure 4 and the commentary on it – the statement that there was a 460% increase does not sound very good. It would be better to normalize the number of ungerminated microconidia under control conditions (without caffeine) to, for example, a value of 1.
Modified accordingly.
- Lines 135, 143 - instead of HaCat it should be HaCaT
Changed to HaCaT as wisely recommended.
- Lines 285-288 - This passage needs to be corrected. First, there is no information on what the suspension was washed with, while the information in parentheses refers to centrifugation conditions. Second, the information about washing and suspending the precipitate in PBS solution is repeated twice.
The methodology description:
“The filtered suspension was washed twice (16,060 x g for 10 min, 4°C) and the pellets resuspended in Phosphate Buffered Saline (PBS) (10 mM Na2PO4, 1.8 mM KH2PO4, 137 mM NaCl, 2.7 mM KCl (w/v), pH 7.3). After another washed step, the pellet was re-suspended in PBS for posterior use for susceptibility assays, in YME (0.4% yeast extract (w/v), 1% glucose (w/v), 1% malt extract (w/v)) for liquid cultures, or in DMEM D5648 (Sigma-Aldrich®, St. Louis, Missouri, USA) supplemented with 10% non-inactivated Fetal Bovine Serum, 10 mM HEPES, 12 mM NaHCO3 and 2 mM L-glutamine for later use in infection assays. Spore concentrations of the purified suspensions were estimated in microconidia/mL using a haemocytometer.”
Was corrected to:
“The filtered suspension was washed twice with Phosphate Buffered Saline (PBS) (10 mM Na2PO4, 1.8 mM KH2PO4, 137 mM NaCl, 2.7 mM KCl (w/v), pH 7.3), by centrifugation at 16,060 x g for 10 min, at 4°C) and the pellets were resuspended in PBS. After another washed step under the same conditions, the pellet was resuspended in PBS and diluted in RPMI R1383 (Sigma-Aldrich®, St. Louis, Missouri, USA) for posterior use for susceptibility assays, was resuspended in YME (0.4% yeast extract (w/v), 1% glucose (w/v), 1% malt extract (w/v)) for liquid cultures, or in DMEM D5648 (Sigma-Aldrich®, St. Louis, Missouri, USA) supplemented with 10% non-inactivated Fetal Bovine Serum, 10 mM HEPES, 12 mM NaHCO3 and 2 mM L-glutamine for later use in infection assays. Spore concentrations of the purified suspensions were estimated in microconidia/mL using a haemocytometer.
- Lines 314-315 - "[31Fernandes et al., 2023]" should be removed.
Corrected accordingly
- Lines 318, 336, 352 - mentagrophytesshould be written in italics.
Changed accordingly